# Tropospheric ozone responses to the El Niño-Southern Oscillation (ENSO): quantification of individual processes and future projections from multiple chemical models

Jingyu Li<sup>1,2,3</sup>, Haolin Wang<sup>1,2,3</sup>, Qi Fan<sup>1,2,3</sup>\*, Xiao Lu<sup>1,2,3</sup>\*

Correspondence to: Xiao Lu (luxiao25@mail.sysu.edu.cn) and Qi Fan (eesfq@mail.sysu.edu.cn)

Abstract. The El Niño-Southern Oscillation (ENSO) modulates tropospheric ozone variability, yet quantitative contributions from individual processes and future response remain unclear. Here, we evaluate the GEOS-Chem chemical transport model and ten Chemistry-Climate Models (CCMs) in Coupled Model Intercomparison Project Phase 6 (CMIP6) in capturing ozone-ENSO response, quantify the roles of transport, chemistry, and biomass burning, and examine the future evolution of these responses. GEOS-Chem simulation over 2005-2020 well reproduces satellite-observed ozone-ENSO response, including the instantaneous decrease (increase) in tropospheric column ozone (TCO) over tropical eastern (western) Pacific in El Niño, and the delayed response in subtropics and mid-latitudes. The combined effects of transport, chemistry, and biomass burning emissions explain over 90% of the simulated TCO variability in tropical Pacific during ENSO. Changes in transport patterns show the dominant role by explaining 53% (+0.8 DU) and 92% (-2.2 DU) of the variability in TCO respectively in the western and eastern Pacific during El Niño relative to normal periods. Chemical depletion reduces ozone by 0.2 and 0.7 DU respectively in the western and eastern Pacific, which is offset by enhanced biomass burning emissions of 0.4 and 0.1 DU. Only five of ten CMIP6 CCMs, with interactive tropospheric chemistry and accurate representation of ENSO dynamics, reproduce the tropical ozone-ENSO response. These models consistently indicate that tropical ozone-ENSO response will increase by 15-40% in 2100 under the SSP3-7.0 scenario, associated with strengthening anomalous circulation and increasing water vapor with global warming. These results are critical for understanding climate-chemistry interactions and for future ozone projection.

<sup>&</sup>lt;sup>1</sup>School of Atmospheric Sciences, Sun Yat-sen University, Southern Marine Science and Engineering Guangdong Laboratory (Zhuhai), Zhuhai, Guangdong 519082, China

<sup>&</sup>lt;sup>2</sup>Guangdong Provincial Observation and Research Station for Climate Environment and Air Quality Change in the Pearl River Estuary, Zhuhai, Guangdong Province 519082, China

<sup>&</sup>lt;sup>3</sup> Key Laboratory of Tropical Atmosphere-Ocean System (Sun Yat-sen University), Ministry of Education, Zhuhai, Guangdong Province 519082, China

# 30 1 Introduction

60

Tropospheric ozone is a greenhouse gas and also plays a crucial role in atmospheric chemistry as the major source of hydroxyl radical (OH) (Lelieveld et al., 2004; Stevenson et al., 2013), the essential oxidant determining the fate of many trace gases including methane. At ground level, it also acts as an air pollutant that affects human health and vegetation growth (Nowroz et al., 2024). Tropospheric ozone is produced chemically from its precursors, including nitrogen oxides (NO<sub>x</sub>=NO+NO<sub>2</sub>), volatile organic compounds (VOCs), and carbon monoxide (CO), with additional source from the stratosphere (Monks et al., 2015). While long-term trends in tropospheric ozone have been shown to be dominated by the shift in the anthropogenic emissions of its precursor (Zhang et al., 2016; Wang et al., 2022), the interannual variability is strongly modulated by climate (Wang et al., 2022). Understanding the impacts of climate variability on ozone, as well as the capability of chemical models to reproduce such effects, becomes increasingly critical with future decrease in anthropogenic emissions of ozone precursors and climate change. Here, we evaluate the ability of multiple chemical models in reproducing the response of tropospheric ozone to the El Niño-Southern Oscillation (ENSO), and quantify the underlying mechanisms, both at the present and under future projections.

ENSO is the primary mode of interannual variability in the equatorial Pacific region (Bjerknes, 1969), and also has a profound impact on weather and climate patterns worldwide (Callahan et al., 2021; Cai et al., 2022). The El Niño (La Niña) conditions are characterized by anomalous increases (decreases) in sea surface temperature in the central and eastern Pacific, which trigger anomalous circulation that weakens (strengthens) the Walker Circulation (Diaz et al., 2001). Previous studies have revealed significant variations in tropical ozone in response to different phases of ENSO through satellite observations. The key response is a notable decrease in tropospheric ozone in the central and eastern Pacific but increase in the western Pacific during the El Niño events, reproducing an east-west "dipole" in tropospheric ozone between the western and eastern Pacific (Chandra et al., 1998; Thompson et al., 2001; Oman et al., 2011; Ziemke et al., 2015). Ziemke et al. (2010) estimated a mean difference of 2.4 (-1.4) Dobson Units (DU) in tropospheric ozone between the western and eastern Pacific in response to 1K change in sea surface temperature in eastern Pacific during El Niño (La Niña) events. A large response of up to 25 DU in tropospheric column ozone was observed over Indonesia during September-November 1997, the period experiencing exceptionally strong El Niño conditions and extreme fires and weather around the world (Page et al., 2002; Picaut et al., 2002), which is comparable to the annual mean level of local tropospheric ozone column (Ziemke and Chandra, 2003). In addition to the impact on tropics, ENSO also modulates tropospheric ozone in the extratropics (Lanford et al., 1998; Balashov et al., 2014; Lin et al., 2015; Olsen et al., 2016; Xu et al., 2017; Jeong et al., 2023). These responses typically lag ENSO by a few months and are weaker compared to tropics (Olsen et al., 2016). The overall impact of ENSO on global tropospheric ozone burden remained unclear through direct observations.

Mechanisms contributing to the ozone-ENSO response have been examined extensively by chemical model simulations. During El Niño, the main processes contributing to ozone decrease in the central and eastern tropical Pacific involve the abnormally upward transport of low-level ozone-poor air and increased ozone chemical loss with enhanced water vapor concentration, while the anomalous subsidence of ozone-rich air and increasing fire activity over Indonesia leads to ozone increase in the western Pacific (Doherty et al., 2006; Lu et al., 2019a). ENSO also modulates tropospheric ozone concentrations by altering tropic lightning NO<sub>X</sub> emissions (Murray et al., 2013), biogenic volatile organic compounds (BVOCs) emissions (Pfannerstill et al., 2018; Vella et al., 2023) and stratospheric-tropospheric exchanges (Doherty et al., 2006; Zeng and Pyle, 2005). A few studies also quantified the relative importance of these processes in shaping the tropical ozone-ENSO response. Sekiya and Sudo (2012) shows that the impact of transport is greater than that of chemistry in contributing to the east-west "dipole" in tropospheric ozone between the western and eastern Pacific. Hou et al. (2016) and Olsen et al. (2016) also support that change in transport is the key process in the ozone-ENSO response. Doherty et al. (2006) shows that increasing biomass burning emissions alone increase tropospheric ozone by 1-2 DU during strong El Niño events. The influence of ENSO on ozone (as well as other trace gases such as CO) in the mid-latitudes are mainly through modulating the dynamics of planetary wave, monsoon, and extratropical jets (Olsen et al., 2016; Lin et al., 2017; Liu et al., 2022; Albers et al., 2022; Yang et al., 2022), and by producing warm weathers inducive for regional ozone chemical production (Xu et al., 2017).

These studies indicate that the response of ozone to ENSO reflects a complicate large-scale coupling between atmospheric chemistry and climate variability. Assessing the accuracy of models in simulating the ozone-ENSO response promotes the improvement of short-term ozone forecast and future ozone projection, particularly for high ozone pollution events that may occur during El Niño. Consequently, the ozone-ENSO response can also serve as a metric for assessing model's ability to capture climate-chemistry interactions. Assessing this metric places higher demands on the models compared to evaluations based solely on simulated concentrations. Previous studies have shown that chemistry-climate models forced with observed or reanalysis sea surface temperatures can reproduce the ozone-ENSO relationship (Doherty et al., 2006; Oman et al., 2011; Sekiya and Sudo, 2012; Hou et al., 2016). However, whether such capability can sustain without constraints from observed sea surface temperature, which are the typical scenarios for future projections, remain unevaluated. It further hinders a robust projection of future ozone-ENSO response using current climate-chemical models. The continuous satellite observations of tropospheric ozone column, as well as the increasing number of chemistry-climate models participating in the Coupled Model Intercomparison Project Phase 6 (CMIP6), now provides a new opportunity to address this issue.

In this study, we apply multiple chemical models to evaluate their ability to reproduce the present-day ozone-ENSO responses, to quantify the contribution from individual process, and to examine the evolution in such response in future scenarios. We start from a 16-year (2005-2020) chemical transport model simulation driven by reanalysis meteorological fields, and compare the simulated ozone-ENSO responses with satellite observations. We then conduct sensitivity simulations to quantify the role of transport, chemistry, and biomass burning in the simulated ozone-ENSO responses. We then use the ozone-ENSO response

90

as a metric to evaluate the ability of ten CMIP6 chemistry-climate models, and rely on the models with successful skills to examine the ozone-ENSO responses in the end of the 21st century under future scenarios.

# 2 Data and Methods

100

105

120

#### 2.1 Tropospheric column ozone measurement from the OMI/MLS instrument

We apply the OMI/MLS product of global tropospheric column ozone (TCO) concentration for 2005-2020 measured from the Ozone Monitoring Instrument (OMI) and Microwave Limb Sounder (MLS) instrument (Ziemke et al., 2006) (https://acdext.gsfc.nasa.gov/Data services/cloud slice/new data.html, last access: 14 January 2025). Both the OMI and MLS instruments fly on NASA's Earth Observing System AURA satellite launched in July 2004. OMI is a nadir-viewing instrument that measures backscattered solar radiance at visible and UV wavelength channels with near global coverage at a nadir footprint resolution of 13km×24km. Total ozone column from OMI is derived from the Total Ozone Mapping Spectrometer (TOMS) version 8 algorithm. The MLS instrument measures vertical profiles of ozone at above the upper troposphere from limb scans ahead of the Aura satellite. TCO is then derived by subtracting measurements of MLS stratospheric column ozone from OMI total column ozone, after adjusting for intercalibration differences of the two instruments using the convective-cloud differential method (Ziemke et al., 2006). Previous study shows an excellent agreement between the TCO from OMI/MLS and those observed from ozonesonde, with a relatively small bias of about 5 DU. These discrepancies mainly arise from from 110 stratosphere-troposphere separation errors (3-5 DU), and cloud contamination (~2 DU) mitigated by filtering scenes with reflectivity >0.3 (Ziemke et al., 2006). The OMI/MLS product provides monthly mean TCO data between 60°S and 60N° at the 1°×1.25° horizontal resolution from October 2004. The product has been widely used in studying global tropospheric ozone distributions (Ziemke et al., 2010; Cooper et al., 2014) and long-term trends (Gaudel et al., 2018; Lu et al., 2019b).

#### 2.2 Niño3.4 index 115

The Niño 3.4 sea surface temperature (SST) anomaly index is one of the most commonly-used indices to inform El Niño and La Niña events. The Niño 3.4 index is provided by the National Oceanic and Atmospheric Administration (NOAA) (https://psl.noaa.gov/gcos/wgsp/Timeseries/Data/nino34.long.anom.data, last access: 14 January 2025) (Rayner et al., 2003). This index is calculated as:

$$Ni\tilde{n}o3.4 \, Index = SST - \overline{SST}_{1981-2010} \tag{1}$$

where SST is the monthly mean SST averaged over the Niño3.4 region (5°N-5°S, 170°W-120°W),  $\overline{SST}_{1981-2010}$  is the 1981-2010 climatological mean SST for the same month over the Niño3.4 region. We use the Niño3.4 index to define the El Niño and La Niña events as will be discussed in Section 2.3.

# 2.3 GEOS-Chem model description and configuration

- We use the GEOS-Chem global chemical transport model (v11-02rc) to interpret the response of tropospheric ozone to ENSO. The model applies the Modern-Era Retrospective analysis for Research and Application version 2 (MERRA-2) data as offline input of meteorological fields. GEOS-Chem includes a detailed mechanism of coupled ozone–NO<sub>x</sub>–VOC–HO<sub>x</sub>–halogen–aerosol to describe stratospheric and tropospheric chemistry (Wang et al., 1998; Eastham et al., 2014). The chemical kinetics are provided by the Jet Propulsion Laboratory (JPL) and International Union of Pure and Applied Chemistry (IUPAC) (IUPAC, 2013). Photolysis rates are calculated by the Fast-JX scheme (Bian and Prather, 2002). Advection of tracers is performed using the TPCORE advection algorithm (Lin and Rood, 1996). Dry deposition for both gas and aerosols is simulated by the resistance-in-series algorithm (Wesely, 1989; Zhang et al., 2001). Wet deposition of water-soluble gas and aerosols is described by Liu et al. (2001) and Amos et al. (2012).
- Emissions used in this study are largely consistent with Wang et al. (2022). We use the Community Emissions Data System inventory (CEDS v-2021-04-21) (McDuffie et al., 2020) for global anthropogenic emissions. Monthly mean biomass burning emissions are from the Global Fire Emissions Database version 4 (GFED4) (van der Werf et al., 2017). The model also includes online calculation of biogenic VOCs (BVOCs) emissions (Guenther et al., 2012), lightning NO<sub>x</sub> emissions (Murray et al., 2012), and soil NO<sub>x</sub> emissions (Hudman et al., 2012; Lu et al., 2021).

We conducted a BASE simulation using year-specific meteorological fields and emission inventory for 2005-2020 with a global horizontal resolution of 2°×2.5° and 72 vertical layers covering the surface to 0.01 hPa. The simulation runs year by year from 1<sup>st</sup> September in the preceding year (the first 4-month as spin-up time), with the initial field provided by a continuous long-term global GEOS-Chem simulation started from 1995 at a horizontal resolution of 4°×5° (Wang et al., 2022).

The application of a chemical transport model (*i.e.*, using offline meteorological fields as input instead of simulating climate online) such as GEOS-Chem (driven by MERRA-2 reanalysis meteorology) allows us to decouple the role of different meteorological parameters in ozone variations. To quantitatively investigate the mechanism of tropospheric ozone response to ENSO, we conducted sensitivity experiments in GEOS-Chem driven by meteorological conditions and emission inventory from specific periods. We first select three periods based on the Niño3.4 index, *i.e.*, the Normal period (January 2013 to December 2013, mean Niño3.4 index of -0.2), the El Niño period (May 2009 to April 2010 and May 2014 to April 2016, mean Niño3.4 index of 1.1), and the La Niña period (July 2007 to June 2008 and June 2010 to May 2012, mean Niño3.4 index of -1.0). These periods cover either one or three complete years, ensuring that ozone contrast in El Niño and La Niña conditions are not influenced by seasonal variations in ozone. For the El Niño and La Niña periods, using three years of meteorological conditions helps minimize potential impacts from other climate modes on ozone. To reduce computational cost, we only choose

year 2013 as the reference year for the Normal period. Evaluation of other climate indexes such as the Indian Ocean Dipole (IOD) index and the Arctic Oscillation (AO) index shows that the year 2013 is not suffered from significant climate variability.

We conducted four sets of sensitivity experiments, as summarized in Table 1. In the TOTAL experiment, meteorological fields affecting atmospheric transport (including three-dimensional wind and pressure fields), affecting chemistry (temperature, water vapor, and cloud cover (which affects radiation)), and biomass burning emissions are all set to conditions in the El Niño and La Niña periods, respectively, while other meteorological conditions and emission inventory remained at 2013 (Normal period) levels. The simulation runs for a three-year period, both for the El Niño and La Niña conditions. The aim of this simulation is to quantify the combined effects of atmospheric transport conditions, chemical conditions, and biomass burning conditions on ozone change during El Niño and La Niña. Similarly, in the TRANSPORT, CHEMISTRY, and BBEMIS simulations, we input the specific meteorological fields and biomass burning emission level under El Niño and La Niña conditions into the model, while keeping other conditions fixed at the 2013 (Normal period) levels. This approach allows us to isolate the impacts of atmospheric transport, chemistry, and biomass burning on ozone under El Niño and La Niña periods.

**Table 1.** Configurations of GEOS-Chem simulations in this study.

|                        | Temperature, water |                         |                                             |                 |  |  |  |
|------------------------|--------------------|-------------------------|---------------------------------------------|-----------------|--|--|--|
| Simulation Name        | Time               | Wind field <sup>a</sup> | vapor, and cloud<br>properties <sup>b</sup> | Biomass Burning |  |  |  |
| BASE <sup>c</sup>      | 2005-2020          | Normal                  | Normal                                      | Normal          |  |  |  |
| $TOTAL^d$              | 2013               | El Niño/La Niña         | El Niño/La Niña                             | El Niño/La Niña |  |  |  |
| TRANSPORT <sup>e</sup> | 2013               | El Niño/La Niña         | Normal                                      | Normal          |  |  |  |
| CHEMISTRY <sup>f</sup> | 2013               | Normal                  | El Niño/La Niña                             | Normal          |  |  |  |
| BBEMIS                 | 2013               | Normal                  | Normal                                      | El Niño/La Niña |  |  |  |

<sup>&</sup>lt;sup>a</sup> The variables include zonal, meridional, and vertical wind fields, and atmospheric pressure.

<sup>&</sup>lt;sup>b</sup> The variables include air temperature, humidity, and cloud properties (which affects radiation).

<sup>&</sup>lt;sup>c</sup> GEOS-Chem runs with yearly specific meteorological fields.

<sup>&</sup>lt;sup>d</sup> GEOS-Chem runs with all meteorological variables of corresponding El Niño/La Niña periods. El Niño/La Niña periods for the simulation are defined in Section 2.3.

<sup>e</sup> GEOS-Chem runs with meteorological variables related to transports (wind and pressure fields) of corresponding El Niño/La Niña periods.

f GEOS-Chem runs with meteorological variables related to chemistry (temperature, humidity, and cloud properties) of corresponding El Niño/La Niña periods.

We note that the TRANSPORT simulation not only changes the spatial distributions of ozone, but also the distributions of ozone precursors such as NO<sub>x</sub> and CO, while it does not change the water vapor concentration in the model which is independently input from the MERRA-2 dataset. This simulation strategy has been applied in previous studies to isolate the impact of transport on ozone variability (Lu et al., 2019b; Kerr et al., 2019). The CHEMISTRY simulation (with only temperature, water vapor, and cloud properties perturbed) not only changes the chemical kinetics, but also modulates the natural emissions such as biogenic VOCs and soil reactive nitrogen which rely significantly on temperature and radiation. The initial conditions for these sensitivity simulations are all obtained from the BASE simulation.

# 2.4 CMIP6 models

An important objective of this study is to evaluate the capability of current cutting-edge chemistry-climate coupled models to reproduce the ozone-ENSO response that profoundly reflects large-scale interactions between climate and chemistry, thereby providing a foundation for robust projection for future ozone-ENSO interactions. Here we obtain output of historical experiment from ten chemistry-climate coupled models participating in CMIP6 (Evring et al., 2016). All the selected CMIP6 models have the same ensemble member "rlilplfl" (except for the UKESM1-0-LL with "rlilplf2"), with "r" representing realization, "i" for initialization, "p" for physics, and "f" for forcing. Table 2 illustrates the differences between the CMIP6 models and GEOS-Chem in the atmospheric dynamics and chemistry modules. Unlike GEOS-Chem which is an offline chemical transport model using archived meteorological fields as input, CMIP6 models perform simulations online using its own atmospheric dynamic module, with externally imposed forcings such as solar variability, volcanic aerosols, and greenhouse gases. In particular, the historical simulations analyzed here do not used observation-based SSTs. Notable difference also exists among the CMIP6 models. The chemical species and schemes vary significantly among the models. For tropospheric ozone chemistry, six models (BCC-ESM1, CESM2-WACCM, EC-Earth3-AerChem, GFDL-ESM4, MRI-ESM2-0, and UKESM1-0-LL) simulate tropospheric chemistry interactively, while four models (AWI-ESM-1-1-LR, IPSL-CM6A-LR-INCA, MPI-ESM-1-2-HAM, and NorESM2-MM) use prescribed tropospheric ozone data. In terms of stratospheric chemistry, four models (CESM2-WACCM, GFDL-ESM4, MRI-ESM2-0 and UKESM1-0-LL) simulate stratospheric ozone chemistry interactively, while BCC-ESM1 and EC-Earth3-AerChem use prescribed climatology values to constrain concentrations of stratospheric species. Table S1 summarizes the ocean components of the CMIP6 models analyzed in this study, including their resolutions. These model configurations represent the current generation of ocean-atmosphere coupling systems used for simulating ENSO dynamics. As a result, the ability of these CMIP6 models to reproduce the observed ozone-ENSO response differs by their atmospheric climate model, chemistry schemes, and the coupling between two. Comparing the results from the offline chemical transport model and online chemistry-climate model provides new insights for understanding the expertise of different types of models.

**Table 2.** Information of the GEOS-Chem and CMIP6 models used in this study.

| Name              | Lat×Lon                                 | Vertical D | Dynamics | Tropospheric ozone | Stratospheric ozone | Reference         |
|-------------------|-----------------------------------------|------------|----------|--------------------|---------------------|-------------------|
|                   |                                         |            |          | chemistry          | chemistry           |                   |
| GEOS-Chem         | 2°×2.5°                                 | L72        | Offline  | Interactive        | Interactive         | Bey et al.        |
|                   |                                         |            |          |                    |                     | (2001)            |
| AWI-ESM-1-1-LR    | ~1.8°×1.8°                              | L47        | Online   | Prescribed         | Prescribed          | Shi et al. (2020) |
| BCC-ESM1          | ~2.8°×2.8°                              | L26        | Online   | Interactive        | Prescribed          | Wu et al. (2020)  |
|                   |                                         |            |          |                    |                     | Danabasoglu et    |
| CESM2-WACCM       | ~0.9°×1.25°                             | L70        | Online   | Interactive        | Interactive         | al. (2020)        |
| EC-Earth3-AerChem | 2°×3°                                   | L34        | Online   | Interactive        | Prescribed          | Döscher et al.    |
|                   |                                         |            |          |                    |                     | (2022)            |
| GFDL-ESM4         | 1°×1.25°                                | L49        | Online   | Interactive        | Interactive         | Dunne et al.      |
| GFDL-ESM4         | 1 ^1.23                                 | L49        | Offinie  | Interactive        |                     | (2020)            |
| IPSL-CM6A-LR-     | ~1.25°×2.5°                             | L79        | Online   | Prescribed         | Prescribed          | Boucher et al.    |
| INCA              |                                         |            |          |                    |                     | (2020)            |
| MPI-ESM-1-2-HAM   | ~1.8°×1.8°                              | L47        | Online   | Prescribed         | Prescribed          | Mauritsen et al.  |
|                   |                                         |            |          |                    |                     | (2019)            |
| MRI-ESM2-0        | ~2.8°×2.8°                              | L80        | Online   | Interactive        | Interactive         | Yukimoto et al.   |
|                   |                                         |            |          |                    |                     | (2019)            |
| NorESM2-MM        | ~0.9°×1.25°                             | L32        | Online   | Prescribed         | Prescribed          | Seland et al.     |
|                   |                                         |            |          |                    |                     | (2020)            |
| UKESM1-0-LL       | ${\sim}1.25^{\circ}{\times}1.8^{\circ}$ | L85        | Online   | Interactive        | Interactive         | Sellar et al.     |
|                   |                                         |            |          |                    |                     | (2019)            |

We select 1980-2014 as the historical scenario (2005-2014 is used for evaluation with OMI/MLS observations), and 2066-215 2100 as the future scenario to compare the present-day and future changes in the ozone-ENSO response. We use the SSP3-7.0 future scenario, representing a medium-emission Shared Socioeconomic Pathway with 7 W/m<sup>2</sup> of additional man-made radiative forcing by the year 2100 (O'Neill et al.,2016; Riahi et al.,2017). This scenario represents one of the most widely used

medium-emission pathways in current climate modeling studies, providing a robust multi-model ensemble for assessing potential ozone-ENSO interactions under anthropogenic climate change. As the CMIP6 simulations analyzed here are not constrained by observed SSTs, the ENSO phases and intensity are not aligned with observations and may differ significantly between models. Following other ENSO-related studies using CMIP6 projections (*e.g.*, Callahan et al., 2021; Cai et al., 2022), we calculate the Niño3.4 index by using the same methodology as Formula 1 but with simulated sea surface temperature (SST) for each CMIP6 model. The Niño3.4 index under future SSP3-7.0 scenario is linearly detrended over 2066-2100. The El Niño and La Niña periods in the CMIP6 models are defined following the NOAA standard, with the Niño3.4 index greater than 0.5 and less than -0.5, respectively.

#### 2.5 Metrics used to quantify the ozone-ENSO response

We mainly use TCO to describe the amount of tropospheric ozone, and the linear correlation between TCO and Niño 3.4 index to quantify the ozone-ENSO response, following several previous studies (Oman et al., 2013; Olsen et al., 2016). For GEOS-Chem and all CMIP6 models, we first convert the simulated gridded ozone concentration in unit of mixing ratio to TCO in unit of DU, using the tropopause from the MERRA-2 reanalysis data. We deseasonalise and detrend the observed and simulated TCO at the corresponding grid to remove the signal of anthropogenic emissions on long-term ozone trends. We report two statistical metrics between the time series of monthly TCO and the Niño3.4 index. First, we report the correlation coefficient between the TCO and the Niño3.4 index (denoted as  $r_{\text{TCO-Niño34}}$ ) at each grid. This coefficient is calculated using the monthly deseasonalised and detrended TCO and Niño 3.4 index, representing the instantaneous response of TCO to ENSO, and also using the TCO time series lagging behind the Niño3.4 index by 3, 6, and 9 months, representing the delayed response of TCO to ENSO. Second, we report the TCO-ENSO sensitivity, denoted as  $m_{\text{TCO-Niño34}}$ , which is the linear regression coefficient of TCO and the Niño3.4 index, quantifying the magnitude of the TCO change in response to a 1K change in the Niño3.4 index. The  $r_{\text{TCO-Niño34}}$  and  $m_{\text{TCO-Niño34}}$  for each grid are calculated as:

$$r_{X-Y} = \frac{\sum_{i=1}^{n} (X_i - \overline{X})(Y_i - \overline{Y})}{\sqrt{\sum_{i=1}^{n} (X_i - \overline{X})^2} \sqrt{\sum_{i=1}^{n} (Y_i - \overline{Y})^2}}$$
(2)

$$m_{X-Y} = \frac{\sum_{i=1}^{n} (X_i - \overline{X})(Y_i - \overline{Y})}{\sum_{i=1}^{n} (Y_i - \overline{Y})^2}$$

$$\tag{3}$$

Where  $X_i$  is the gridded monthly deseasonalised and detrended TCO,  $Y_i$  is the monthly Niño3.4 index. These metrics effectively normalize comparisons across models with differing climate variability backgrounds. We report the p-value of corresponding  $r_{\text{TCO-Niño34}}$  and  $m_{\text{TCO-Niño34}}$  where applicable, but we do not use thresholds such as  $p \le 0.05$  to judge whether the reported values are statistically significant, as advised by the statistics community (Wasserstein et al., 2019). Still, smaller p-value indicates higher statistical confidence.

# 3 The ozone-ENSO response and contribution from individual processes

# 3.1 The responses of tropospheric ozone to ENSO from satellite observations and GEOS-Chem model simulations

Figure 1 compares the spatial and seasonal distributions of TCO concentrations from the OMI/MLS satellite product and 250 GEOS-Chem simulation, averaged over 2005-2020. Observations from OMI/MLS indicate high tropospheric ozone levels in the mid-northern latitudes during the boreal spring and summer, and in the southern Atlantic and southern Africa in boreal autumn. This spatiotemporal distribution can be attributed to the stronger stratosphere-troposphere exchange in springtime in both hemispheres, the higher photochemical production in the mid-northern latitudes during the boreal summer, and biomass burning emissions in Africa in boreal autumn (Cooper et al., 2014). The GEOS-Chem model well captures these observed 255 TCO features, with high spatial correlation coefficients ranging from 0.70 to 0.86 for all seasons and a small seasonal mean bias of -2.1 to -1.3 DU relative to OMI/MLS observations. Larger discrepancies are shown at high latitudes in both hemispheres, where satellite retrievals of tropospheric ozone may have lager uncertainty due to high solar zenith angle and low surface albedo conditions. In the tropics, where ENSO has the largest influences on ozone, GEOS-Chem model results are in excellent agreement with the OMI/MLS products. The model also shows a good consistency with the observed monthly variation of TCO averaged over 60°S-60°N (Figure S1), but underestimates the long-term trends in TCO (0.07 DU year-1 in GEOS-Chem 260 vs 0.15 DU year-1 in OMI/MLS). This underestimation of long-term trends has small impacts on the following analysis as long-term trends in TCO will be removed before examining the ozone-ENSO responses.

Figure 1. Spatial and seasonal distributions of tropospheric ozone from (a) OMI/MLS satellite observations, (b) GEOS-Chem simulation, and (c) differences between the two (model results minus observations) with the seasonal mean differences (±standard deviations) shown inset. Values are 16-year averages over 2005-2020. The black box indicates the Niño3.4 region.

Figure 2 and Figure S2 shows the spatial distributions of the  $r_{\text{TCO-Niño34}}$  and  $m_{\text{TCO-Niño34}}$  derived from the monthly TCO and Niño3.4 index over 2005-2020, respectively. Robust positive (negative)  $r_{\text{TCO-Niño34}}$  is found in the western (eastern) tropical Pacific region, indicating that during the El Niño period, tropospheric ozone levels increase in the western Pacific region while decrease in the Eastern Pacific (Fig.2a). We define two key regions in the western Pacific (WP, 80°-140°E, 20°S-20°N) and the central and eastern Pacific (EP, 170°E-80°W, 20°S to 20°N) showing a significant but contrasting ozone response to ENSO. The selection of these regions is based on consistency with the Ozone ENSO Index (OEI) framework (Ziemke et al., 2010), which also ensures that ~90% of the grid points within the regions shows consistent  $r_{\text{TCO-Niño34}}$  with a low p-value. The WP region shows a regional mean  $r_{\text{TCO-Niño34}}$  of 0.45 and  $m_{\text{TCO-Niño34}}$  of 1.3 DU K-1, with a maximal value of 0.62 and 2.2 DU K-1, respectively. The TCO in the EP shows a slightly stronger response to ENSO compared to the WP, with a regional mean  $r_{\text{TCO-Niño34}}$  of -0.49 and  $m_{\text{TCO-Niño34}}$  is -1.3 DU K-1, and a minimum value of -0.78 and is -2.3 DU K-1, respectively. We also see from Figure 2 that areas showing negative  $r_{\text{TCO-Niño34}}$  are larger than those showing positive  $r_{\text{TCO-Niño34}}$ . The reported  $r_{\text{TCO-Niño34}}$  and observations in earlier periods (Sekiya et al., 2012; Hou et al., 2016). Besides the responses in the tropical Pacific, we also find

positive  $r_{\text{TCO-Niño34}}$  over the Southern Pacific around 30°S, central Africa, northern South America, and negative  $r_{\text{TCO-Niño34}}$  over central South America.

**Figure 2.** The spatial distributions of the  $r_{\text{TCO-Niño}34}$  (lagged correlation coefficients between Niño3.4 index and TCO which is deseasonalised and detrended over 2005-2020) from OMI satellite observations and GEOS-Chem results. Dots denote where the correlation coefficients are with p>0.05, but we note that we do not use thresholds such as p 

Figure 3. The difference in the deseasonalised and detrended TCO between WP and EP region over 2005-2020 from OMI (black lines) and GEOS-Chem simulation (purple lines). The Niño3.4 Index (blue dash lines) is also shown. The grey, red and blue shading in the figure represent the Normal (year 2013), El Niño, and La Niña periods, respectively. The El Niño and La Niña periods are defined as consecutive 1-2 years with the Niño3.4 index greater than 0.5 and less than -0.5 (Yeh et al., 2022), selected for sensitivity experiments in this study, respectively.

The above analysis focuses on instantaneous response of TCO to ENSO. The lagged response of tropospheric ozone to ENSO may provide a potential indicator for ozone prediction, yet it is much less explored compared to the instantaneous ozone-ENSO response. We use the lagged  $r_{\text{TCO-Niño34}}$  (with Niño3.4 index leading TCO by 3, 6, 9 months) to quantify the delayed response of TCO to ENSO, as shown in Figures 2c,e,g. Results show that, three months after changes in SST in the Niño3.4 region, the WP and EP regions remain as the key areas showing the most significant ozone-ENSO response, but both  $r_{\text{TCO-Niño34}}$  and  $m_{\text{TCO-Niño34}}$  weaken, with the mean  $r_{\text{TCO-Niño34}}$  falling to 0.35 and -0.43 in WP and EP regions, respectively. In the meantime, negative  $r_{\text{TCO-Niño34}}$  remains in the central South America and develops over eastern Asia, India, and Middle East, while positive  $r_{\text{TCO-Niño34}}$  extends in the Pacific-North America (black box in Fig.2c). For the lagged six and nine months cases, the  $|r_{\text{TCO-Niño34}}|$ 

decrease significantly over the tropical Pacific, while the positive  $r_{TCO-Niño34}$  develops in the Africa and South America (black box in Fig.2g), indicating notable TCO increase in these regions after six or nine months after the El Niño events.

We find that the GEOS-Chem chemical transport model largely reproduces the instantaneous and delayed response of tropospheric ozone to ENSO (Fig.2b). The simulated regional mean  $r_{\text{TCO-Niño34}}$  over the WP and EP regions are 0.40 and -0.49 over 2005-2020, align closely with the observed values of 0.45 and -0.49. The simulated TCO-ENSO sensitivities ( $m_{\text{TCO-Niño34}}$ ) are 1.2 and -1.5 DU K<sup>-1</sup>, which also agree well with the observed values of 1.3 and -1.3 DU K<sup>-1</sup>. The timeseries of monthly TCO difference between the WP and EP region is also in excellent agreement with the OMI/MLS observations, yielding a consistent correlation coefficient between the TCO differences and the Niño3.4 index (0.75 for GEOS-Chem versus 0.74 for OMI/MLS product) (Fig.3). The model also reproduces the positive  $r_{\text{TCO-Niño34}}$  over the Southern Pacific, central Africa, and the negative  $r_{\text{TCO-Niño34}}$  over central South America (Fig.2b). In terms of the delayed response, we find that GEOS-Chem model captures the weakened ozone-ENSO response over tropical Pacific after 3-9 months with changes in SST in the Niño3.4 region, the sustained  $r_{\text{TCO-Niño34}}$  over central Africa, and the shift from negative to positive  $r_{\text{TCO-Niño34}}$  over South America. The above analyses illustrate that, with constraints from reanalysis meteorological fields, chemical transport model can successfully capture the observed ozone-ENSO relationship. Responses of simulated surface ozone concentrations are mostly similar those of TCO (Fig.S3), including the east-west "dipole" over tropical Pacific, and a notable surface ozone enhancement over South America in 6-9 months lagged by increase in SSTs in the Niño3.4 region.

We also examine the response of tropospheric ozone burden over a larger scale (60°S-60°N) (Fig.S4). On the interannual scale, the OMI/MLS product shows a positive correlation coefficient (r=0.38) between the TCO averaged over 60°S-60°N and the Niño3.4 index over 2005-2020, but such response is not captured by the GEOS-Chem model. However, both the OMI/MLS product and the GEOS-Chem simulation show positive correlation (r=0.42 and r=0.58, respectively) between the TCO averaged over 60°S-60°N and the Niño3.4 index from one-year earlier. Such positive correlation holds for GEOS-Chem when global (i.e. 90°S-90°N) tropospheric ozone burden is calculated (r=0.56). These results indicate that the positive (negative) phase of ENSO increase (decrease) global tropospheric ozone burden in the next year. Our analysis is consistent with Zeng and Pyle (2005), which reported large increase of tropospheric ozone burden following the 1997-1998 El Niño year attributable to increasing stratosphere-troposphere exchange. On the monthly scale, the OMI/MLS product shows only weak correlation between TCO averaged over 60°S-60°N and the Niño3.4 index (r=-0.09), indicating that on a short time scale, ENSO mostly influence the spatial distributions of tropospheric ozone rather than its total amount.

### 3.2 Quantitative contribution from individual processes to the ozone-ENSO response

We now quantify the response mechanism of tropospheric ozone to ENSO by the GEOS-Chem sensitivity simulations. Figure
4 presents the horizontal distribution of TCO changes due to the combined and individual effects of transport, chemistry, and
biomass burning emissions, by contrasting the model results using El Niño/La Niña conditions with the Normal periods as

defined in Section 2.3. Figure 5 displays the vertical distribution of ozone change in the equatorial region (5°S-5°N). Table 3 summarizes the response over the WP and EP region.

**Figure 4.** Distribution of tropospheric ozone column (TCO) concentrations caused by different pathways during El Niño (left panels) and La Niña (right panels). The top panel (a-b) shows the TCO difference between the El Niño (La Niña) periods and Normal periods in the BASE simulation. The following panels quantify the TCO differences caused by the combined and individual contributions from transport, chemistry, and biomass burning, estimated from the corresponding sensitivity experiments (Table 1).

**Figure 5.** Similar to Figure 4, but showing the vertical distribution of ozone concentration in the equatorial region (5°S-5°N). Panels e-f overlay the difference in wind fields between the El Niño (La Niña) periods and the Normal periods. Panels g-h overlays the difference in

water vapor concentration in unit of 10<sup>5</sup> ppbv. The black solid line represents the tropopause. The gray dash lines outline the WP and EP regions.

360

370

375

**Table 3.** TCO changes due to the combined and individual effects of transport, chemistry, biomass burning emissions and interactive effect.

| TCO difference IDIII8              | El N | Niño | La N | La Niña |  |
|------------------------------------|------|------|------|---------|--|
| TCO difference [DU] <sup>a</sup> _ | WP   | EP   | WP   | EP      |  |
| BASE simulation                    | 1.6  | -2.4 | -0.7 | 0.9     |  |
| Combined effect                    | 1.5  | -2.4 | -0.5 | 1.1     |  |
| Transport                          | 0.8  | -2.2 | -0.6 | 0.8     |  |
| Chemistry                          | -0.2 | -0.7 | -0.6 | -0.2    |  |
| Biomass burning emissions          | 0.4  | 0.1  | 0.1  | 0.1     |  |
| Interactive effect <sup>b</sup>    | 0.5  | 0.4  | 0.6  | 0.4     |  |

<sup>&</sup>lt;sup>a</sup> Values are estimated by contrasting the model results using El Niño/La Niña conditions with the Normal periods over the WP and EP region from the sensitivity simulations.

Our model simulations estimate that during the El Niño periods, the combined effects of transport, chemistry, and biomass burning emissions increase TCO by 1.5 DU averaged over the WP region, and decrease TCO by 2.4 DU in the EP region relative to the Normal period, respectively (Fig.4c). These ozone differences contribute to 94% and 98% of the TCO difference between El Niño and Normal periods in the BASE simulation in the WP and EP regions (Fig.4a, Table 3), respectively, indicating that these three processes investigated in this study are major mechanisms explaining the ozone-ENSO response. These ozone changes also account for 10~20% of the annual mean TCO concentrations in the WP and EP regions, highlighting the large influence of ENSO on the interannual variability of tropical tropospheric ozone. In comparison, the TCO changes are smaller in the La Niña condition, showing a decrease in TCO of 0.7 DU and an increase in TCO of 0.9 DU averaged for the WP and EP region, respectively, compared to the Normal conditions. The ozone response to ENSO also exhibits significant vertical variations (Fig.5), with the greatest ozone change occurring in the upper troposphere. For instance, during the El Niño

<sup>365</sup> b The interactive effect is derived as the difference between the combined effect and the additive effect of transport, chemistry, and biomass burning emissions.

conditions, the maximum ozone change is found near 100 hPa reaching 11 and -40 ppbv respectively in the WP and EP regions, relative to Normal periods.

Both Figure 4 and Figure 5 illustrate that change in atmospheric transport is the dominant factor contributing to the overall tropospheric ozone-ENSO response. Changes in transport patterns alone increase TCO by 0.8 DU in WP region and decrease TCO by 2.2 DU in the EP region during the El Niño condition relative to the Normal periods, accounting for 53% and 92% of the combined effects on TCO change, respectively. As for La Niña, changes in transport patterns alone decrease (increase) TCO by 0.6 (0.8) DU in the WP and EP region, respectively, accounting for 86% and 89% of the combined effect (Table 3). Transport also dominates the spatial patterns of ozone change during El Niño and La Niña conditions compared to the Normal periods, including the responses outside the tropical Pacific (Figure 5). Sekiya et al. (2012) similarly found that the impact of transport on tropospheric ozone outweighs that of chemistry over most of the globe during ENSO. The SST conditions during the El Niño phase boost an abnormal Walker Circulation, featuring abnormal uplifts in the EP region and subsidence in the WP region. As ozone concentrations rise with altitude, abnormal downdraft in the WP region transports ozone-rich air in the upper layer downward, leading to ozone increase extending from tropopause to the surface, while the abnormal updraft lifts ozone-poor air over the ocean to the upper troposphere in the EP, leading to decrease in TCO (Fig.5e,f). We also find that the transport-induced ozone changes are larger in the upper troposphere. In the troposphere above 500 hPa, changes in transport pattern alone explain 85-90% of the simulated ozone increases in the WP region during the El Niño condition, and 67-88% in EP region during La Niña.

Chemical processes also contribute to the ozone-ENSO response. Our GEOS-Chem simulation shows that, nudging atmospheric temperature, water vapor, and cloud cover from the El Niño conditions can cause an averaged TCO change by 0.2 DU in WP region and 0.7 DU in the EP region relative to Normal period, respectively (Fig.4g). These parameters modulate the chemical kinetics, in particular the H<sub>2</sub>O-induced chemical loss of ozone, and natural emissions of BVOCs and soil NO<sub>x</sub> emissions. The largest ozone change induced by these chemical responses are shown in the lower and middle troposphere (500-800 hPa) in the EP region, leading to an average ozone decline of 2.0 ppbv (Fig.5g), which is more than half of the transport-induced ozone decrease of 3.8 ppbv there. This chemistry-induced negative ozone response in the EP region during the El Niño events can be attributed to the increase in water vapor, which chemically depletes tropospheric ozone in the low-NO<sub>x</sub> marine environment. Figure 5g shows that during the El Niño periods, the water vapor over the EP region increases by 9×10<sup>5</sup> ppbv (10%) on average compared to the Normal periods. In comparison, ozone changes induced by lightning NO<sub>x</sub> and BVOCs (e.g., isoprene) in response to temperature variations are smaller than those driven by water vapor in the CHEMISTY experiment. Our model simulation yields a small decrease in lightning NO<sub>x</sub> emissions of 6% an increase in biogenic isoprene emissions of 4% during the El Niño conditions in the WP region. In the EP region, both lightning NO<sub>x</sub> and biogenic isoprene emissions exhibit negligible changes. Thus, the chemistry effect is dominated by changes in water vapor, especially on the global scale.

While the overall direction of the chemistry-induced ozone response during the El Niño events is mostly consistent with Sekiya et al. (2012), one notable difference is that our simulation shows a much smaller positive chemistry-induced ozone increase in the lower troposphere over the WP region, compared to the results shown in Sekiya et al. (2012). This is likely attributed to the different way to separate the role of transport, chemistry, and emissions between the two studies. Sekiya et al. (2012) examined the transport effects by conducting a chemical model simulation with fixed the chemical field at 1990 level, and attribute the remaining ozone response to chemical impact. As the chemical production and loss of ozone are fixed in the simulation, the transport-induced changes in ozone precursor do not translate into changes in ozone chemical production. As a result, these transport-induced ozone changes are not quantified as the transport effect in Sekiya et al. (2012), but are instead attributed to chemistry. In contrast, our simulation strategy attributes the ozone changes induced by transport of ozone precursors to the transport effect. We indeed find that the changes in transport patterns during the El Niño conditions also increase (decrease) the concentrations of NO<sub>x</sub> and peroxyacetyl nitrate in the WP (EP) region, which deepens the east-west "dipole" in tropospheric ozone (Fig.S5). While the comparison with Sekiya et al. (2012) highlights the complex interaction between atmospheric chemistry and transport that may complicate the quantitative attribution of these effects on ozone change, both studies concur that the influence of chemical processes on ozone response to ENSO is less significant than that of transport despite the use of different models and attribution strategy.

We separately quantify the contribution of biomass burning emissions in the ozone-ENSO response. Biomass burning emissions alone increase TCO by 0.4 DU in the WP region during El Niño period, accounting for 27% of the total TCO change. The enhanced ozone due to biomass burning emissions are more pronounced in the middle and lower troposphere over the WP region (Figure 5i), contributing to an average ozone increase of 1.2 ppbv at the surface in the El Niño conditions relative to Normal periods. This is attributed to the warmer and drier weather conditions in the WP region during El Niño periods, which triggers larger biomass burning emissions (Xue et al., 2021). The GFED inventory applied in our simulation indicates that biomass burning emissions of CO in the WP region are 34-63 Tg (790-1465%) larger in the El Niño than the Normal period (Fig.S6c). In the extreme 1997/1998 events, the mean biomass burning emissions for August-October on 1997 are nearly 50 times as the 2005-2020 climatology. Chandra et al. (2002) estimated an increase in TCO by 12–16 DU over Indonesia due to biomass burning in October 1997. In the La Niña conditions, the larger biomass burning emissions contribute to ozone increase over the Brazilian plateau by 0.7 DU, compared to the Normal periods (Fig.4j), which is also supported by the GFED inventory.

We can quantify the interactive effect between the chemistry, transport, and biomass burning emissions as the TCO difference between the combined effect and the additive values from the individual effects. Results are shown in Table 3. We find that for the WP region during the El Niño period and the EP region during the La Niña period, the interactive effect tends to amplify the ozone increase. Conversely, for the WP region during La Niña period and the EP region during the El Niño period, the

interactive effect tends to weaken the ozone decrease. Part of this interactive effect can be clearly illustrated, as can be seen from the comparison of our result to Sekiya and Sudo (2012) as discussed above. For example, during the El Niño period, higher surface temperature over Indonesia due to the anomalous subsidence may further amplify ozone production from biomass burning emissions, thus the interactive effect leads to a further ozone increase. However, quantifying each interactive mechanism requires much more additional model experiments with more complicated design. Nevertheless, the above analysis again highlights the complex interaction between natural sources, chemistry, and transport in the ozone response to climate variability (Lu et al., 2019a).

The lagged ozone-ENSO responses over extratropics involve multiple mechanisms with strong regional variability. Here we discuss the possible mechanisms contributing to the increase in tropospheric ozone in the Pacific-North America 3 months lagged behind Niño3.4 index (as indicated by positive  $r_{\text{TCO-Niño34}}$  in Fig.2c,d) and that in Africa and South America 9 months lagged behind Niño3.4 index (Fig.2g,h), as these two features are prominent in both the OMI/MLS observations and GEOS-Chem simulations. Previous studies have shown increased tropospheric ozone over Pacific–North America in about 3 months lagged after the wintertime El Niño, which may be attributed to the shifts in subtropical jet stream that enhances long-range transport of Asian pollution and biomass burning plumes (Lin et al., 2014; Xue et al., 2021). Our base simulation reproduces the positive  $r_{\text{TCO-Niño34}}$  in the Pacific-North America 3 months lagged behind Niño3.4 index, and attribute the response to changes in transport patterns. The increase in tropospheric ozone in Africa and Brazil 9 months lagged behind Niño3.4 index are likely attributed to enhanced biomass burning emissions (Fig.S6d) (Cordero et al., 2024).

#### 4 The ozone-ENSO relationship in CMIP6 models and its future changes

450

#### 4.1 Comparison of ozone-ENSO relationship in GEOS-Chem and CMIP6 models

Section 3 illustrates the capability of the GEOS-Chem chemical transport model (CTM) to reproduce the ozone-ENSO response. This capability largely results from the use of reanalysis meteorological fields as inputs to drive the CTM, which reproduce the response of atmospheric transport patterns to ENSO events accurately. In contrast, coupled Chemistry-Climate Models (CCMs) utilize their own atmospheric models to drive the chemistry simulations. For the historical experiment in CMIP6, these simulations are not constrained by observed sea surface temperatures. This introduces an additional challenge in simulating the ozone-ENSO response. Since CCMs are indispensable tools for future projections of atmospheric chemistry and air quality, it is crucial to evaluate their ability to capture climate-chemistry interactions. In this section, we demonstrate that the sensitivity of ozone to ENSO can serve as a valuable indicator for this purpose. We focus on the performance of ten CCMs included in CMIP6 for capturing the present-day ozone-ENSO response, and examine how this relationship changes in future scenarios using models with successful skills.

Figure 6. The  $r_{\text{TCO-Niño34}}$  over 2005-2014 from OMI/MLS satellite observations, GEOS-Chem and ten CMIP6 models results. Dots denote regions showing  $r_{\text{TCO-Niño34}}$  with p-values>0.05

Figure 6 and Figure S7 compare the spatial distribution of  $r_{\text{TCO-Niño34}}$  and  $m_{\text{TCO-Niño34}}$  from OMI satellite observations, GEOS-Chem, and ten CCMs for overlapping years of 2005-2014. Figures 7a and 7e summarizes the comparison of  $r_{\text{TCO-Niño34}}$  from

different models to the OMI/MLS observations in the Taylor diagram, respectively in the WP and EP regions. We note here that the Niño3.4 index and the ENSO events from each model is derived from their own simulated SSTs, following the NOAA procedures as described in Section 2.2. This ensures that the ozone-ENSO relationship derived from the CMIP6 models reflects the interactions simulated by the model and is not influenced by the very likely bias in simulated SSTs relative to the observation. We apply the following metrics, including the mean, the maximum (for WP) or minimum (for EP region)  $r_{\text{TCO-Niño34}}$  and  $m_{\text{TCO-Niño34}}$  values, and the proportion of area showing positive (for WP) or negative (for EP region)  $r_{\text{TCO-Niño34}}$  in the EP and WP regions, to quantitatively evaluate the ability of these models. Results are summarized in Figure 8 and Table S2.

**Figure 7.** Comparison of *r*<sub>TCO-Niño34</sub> from OMI satellite observations and model simulations, as well as wind fields from MERRA-2 reanalysis data and model results over 2005-2014. Results are illustrated in the Taylor diagram. Only six CMIP6 models that can capture the overall tropical ozone-ENSO responses are included. For Panels (a) and (e), the reference *r*<sub>TCO-Niño34</sub> is from the OMI/MLS observations. For other panels, the reference data is the MERRA-2 reanalysis fields used from GEOS-Chem simulation. The radian axis represents the spatial correlation coefficient between model result and reference data, the X-axis and Y-axis represent the standard deviation (normalized to observations). The distance between a model marker and the reference point ("REF") quantifies the root-mean-square error (RMSE). Thus, markers closer to "REF" indicate better overall performance. By design, the result would not be presented if the simulated values show negative correlation coefficient with the reference data. The comparison is conducted separated for the WP (top panels) and EP (bottom panels) regions. All results are re-grided to a spatial resolution of 2.8°×2.8° (the resolution from the BCC-ESM model) for comparison.

**Figure 8.** Comparison of GEOS-Chem and different CMIP6 models in capturing the tropical ozone-ENSO relationship. The comparison is conducted separately for the WP (left panels) and EP (right panels) regions. The metrics used in the comparisons are the mean ( $|r|_{\text{mean}}$ ,  $|m|_{\text{mean}}$ ), the maximum (for WP) or minimum (for EP region)  $r_{\text{TCO-Niño34}}$  and  $m_{\text{TCO-Niño34}}$  (illustrated in absolute values  $|r|_{\text{max}}$ ,  $|m|_{\text{max}}$ ), and the proportion of area (Area%) showing positive (for WP) or negative (for EP region)  $r_{\text{TCO-Niño34}}$  in the EP and WP regions.

Our analysis reveals significant discrepancies among CMIP6 models in reproducing the ozone-ENSO responses. The ability to capture the ozone-ENSO relationship largely depends on whether these models include interactive chemistry. Five of the ten CCMs, including BCC-ESM1, CESM2-WACCM, EC-Earth3-AerChem, GFDL-ESM4, and MRI-ESM-2-0, shows  $r_{\rm mean}$ of 0.38±0.07 (mean±standard deviation) and -0.57±0.08 over the WP and EP region, respectively, with 85±14% and 83±9% of the areas showing positive and negative  $r_{\text{TCO-Niño}34}$ . These numbers are closed to the OMI/MLS observations ( $r_{\text{mean}}=0.42$ with 94% showing positive  $r_{\text{TCO-Niño34}}$  in the WP region,  $r_{\text{mean}}$ =-0.52 with 89% showing negative  $r_{\text{TCO-Niño34}}$  in the EP region) as well as the GEOS-Chem simulation for the same period. These five models show  $m_{\rm mean}$  of 1.0±0.4 DU K<sup>-1</sup> for the WP region and -1.4±0.3 DU K<sup>-1</sup> for the EP region, compared to the observed values of 1.3 and -1.4 DU K<sup>-1</sup>, respectively. These models all include interactive tropospheric chemistry. We also find that these models typically show better agreement to the observations in the EP than the WP region, as the reported values of most metrics are less scattered in the EP region (Fig.8). The UKESM1-0-LL model also interactively describes tropospheric chemistry and captures the negative  $r_{\text{TCO-Niño34}}$  in the EP, but it fails to reproduce the response in the WP. In contrast, other four models with prescribed or simplified tropospheric chemistry, including the AWI-ESM-1-1-LR, IPSL-CM6A-LR-INCA, MPI-ESM-1-2-HAM, and NorESM2-MM models, simulate only weak or even contrast response of tropospheric ozone to ENSO. Our study highlights that interactive tropospheric chemistry mechanisms are essential prerequisites for accurately simulating the ozone-ENSO response in CCMs. This is also consistent with the findings from Le et al. (2024).

Section 3 has pointed out the key role of atmospheric transport in shaping the ozone-ENSO response. We examine here whether differences in the simulated changes in transport pattern can explain the variation in ozone-ENSO response among the CCMs. Here we focus on the six CCMs with interactive tropospheric ozone chemistry. Figure 7 compares the vertical and horizontal wind fields from the CCM simulations to the MERRA-2 reanalysis data used in GEOS-Chem as a standard reference during the ENSO events over the EP and WP regions, respectively. We find that the ability to reproduce the abnormally vertical 530 transport in the EP and WP region during the ENSO events is crucial for reproducing the ozone-ENSO response. This is evident in Figure 7b and 7f. Five models (BCC-ESM1, CESM2-WACCM, EC-Earth3-AerChem, GFDL-ESM4, and MRI-ESM-2-0) with successful skills to reproduce the ozone-ENSO response all captures the overall response of vertical velocity to ENSO. The vertical velocity simulated by the BCC-ESM1 model shows relatively large differences compared to MERRA-2 and other models. These discrepancies are also reflected in Table S2, which indicates that the BCC-ESM simulation of  $r_{\text{TCO}}$ . 535 Niño34 has a larger deviation from observations compared to other models. In contrast to these five models, UKESM1-0-LL model fails to reproduce the response of vertical velocity to ENSO over the WP region, which is the most likely reason leading to its inability to capture the positive  $r_{\text{TCO-Niño}34}$  (Fig. 61). The simulation of horizontal winds also shows appreciable differences compared to the MERRA-2 reanalysis data and among different models, but the impact on the  $r_{TCO-Niño34}$  performance is relatively small. It further confirms the key role of atmospheric transport in shaping the ozone-ENSO response.

The analyses above indicate that a number of CMIP6 models can now accurately reproduce the observed ozone-ENSO response, without constraints of observed SSTs. The common feature of these successful models is their inclusion of interactive tropospheric chemistry, and the ability to capture the response of atmospheric circulation to sea surface temperature anomalies during ENSO events, especially the vertical circulation. These capabilities are crucial for using these models to study large-scale climate-chemistry interactions.

# 4.2 The future changes of ozone-ENSO in CMIP6 models under SSP3-7.0 scenario

We now apply CCMs in CMIP6 with successful skills to explore the changes in ozone-ENSO responses in the end of the 21st century under SSP3-7.0 scenario in 2066-2100. We use four CCMs, including CESM2-WACCM, EC-Earth3-AerChem, GFDL-ESM4, and MRI-ESM2-0. We do not include BCC-ESM because its future projection only extends to 2050.

**Figure 9.** The  $r_{\text{TCO-Niño}34}$  in (a) historical (1980-2014) and (b) future (2066-2100) periods under the SSP3-7.0 scenario from four CMIP6 models with available future projections. Dots denote regions showing  $r_{\text{TCO-Niño}34}$  with p-values>0.05.

Figure 9 compares the spatial distributions of  $r_{\text{TCO-Niño34}}$  in present-day (1980-2014) and future (2066-2100) periods as projected by the four models. All four models consistently indicate that the key response of tropospheric ozone to ENSO, featured by the "dipole" pattern of  $r_{\text{TCO-Niño34}}$  in the tropics, remains in the future SSP3-7.0 scenario. In addition, these models project an increased  $|r_{\text{TCO-Niño34}}|$  over both the WP and EP regions, with expanding areas showing  $|r_{\text{TCO-Niño34}}|$  with a p-value

**Figure 10.** Future intensification of tropical ozone-ENSO response projected by the CMIP6 models. The figure compares the mean ( $|r|_{\text{mean}}$ ,  $|m|_{\text{mean}}$ ), the maximum (for WP) or minimum (for EP region)  $r_{\text{TCO-Niño34}}$  and  $m_{\text{TCO-Niño34}}$  (illustrated in absolute values  $|r|_{\text{max}}$ ,  $|m|_{\text{max}}$ ), and the

proportion of area (Area%) showing positive (for WP) or negative (for EP region)  $r_{\text{TCO-Niño34}}$  in the EP and WP regions for historical (1980-2014) and future (2066-2100) SSP3-7.0 scenario, from four CMIP6 models with available future projections. The comparison is conducted separately for the WP (left panels) and EP (right panels) regions, and for  $r_{\text{TCO-Niño34}}$ , Area% (top panels) and  $m_{\text{TCO-Niño34}}$  (bottom panels). Number counts of metrics showing increase in future compared to historical periods from the models are shown inset.

We examine the possible mechanisms contributing to the enhanced ozone-ENSO response over the tropics in the future SSP3-7.0 scenario. Figure 11 shows the difference in concentrations of ozone and water vapor, as well as transport patterns over the tropical regions (averaged over 5°S-5°N), between the positive and negative phases of ENSO, simulated by different CMIP6 models in the present-day level and under SSP3-7.0 scenarios. In the WP region, three models (CESM2-WACCM, EC-Earth3-AerChem, and MRI-ESM2-0) predict increase in tropospheric ozone concentration by 1-9 ppbv from the present-day level and to the future. This ozone enhancement is associated with intensified anomalous downdrafts by 0.02-0.06 pa/s (50-150%), which allows ozone-rich air transport from the upper layers. The GFDL-ESM4 model predicts a slight reduction of 0.5 ppbv in ozone due to a weakened anomalous downdraft of 0.01 pa/s. For the EP region, the models show stronger agreement. All four models predict that the negative ozone difference between the El Niño and La Niña phases will intensify by about 6-12 ppbv. This also partly reflects the stronger anomalous updraft between the El Niño and La Niña phases in the future than the present-day level. In addition, the warming future features higher water vapor in the atmosphere, which results in stronger ozone chemical depletion during the El Niño phases in the EP region. The CESM is the only model providing fields of water vapor concentration. We find that the global mean concentrations of water vapor increase by 28% in 2066-2100 compared to 1980-2014, and that the east-west "dipole" pattern water vapor anomaly deepens. Overall, the stronger ozone-ENSO response in the future SSP3-7.0 scenario is a robust feature predicted by all models with successful skills in reproducing historical ozone-ENSO response, and reflects a combined effects of dynamics and chemistry.

**Figure 11.** Future intensification of tropical ozone-ENSO response projected by the CMIP6 models. Left and middle panels show the ozone difference between the El Niño and La Niña conditions in the equatorial region for the historical and future periods, respectively. Right panels show the difference between future and historical periods. Difference in wind fields and water vapor concentration are overlaid (water vapor concentrations for future scenarios are only available from the CESM-WACCM2 model so are only shown in panel c).

#### 5 Discussion and Conclusion

In this study, we integrate satellite observations with multiple global chemical models to examine the response of tropospheric ozone to ENSO, both in present-day and under future scenarios. We aim to provide a comprehensive evaluation of the capability of state-of-the-art chemical models in reproducing the ozone-ENSO response, quantify the individual processes contributing to this large-scale response of atmospheric chemistry to climate variability, and from which to deepen our understanding of climate-chemistry interactions and to improve future ozone projections.

The GEOS-Chem chemical transport model, driven by prescribed MERRA-2 reanalysis meteorological fields, show excellent skills in reproducing the key ozone-ENSO response, featured by the notable decrease in tropospheric ozone in the central and eastern Pacific and the increase in the eastern Pacific during the El Niño. It shows good agreement with the OMI/MLS satellite products in terms of the regional mean  $r_{\text{TCO-Niño34}}$  and  $m_{\text{TCO-Niño34}}$  over the WP and EP regions in 2005-2020, and also captures ozone changes in the subtropics and mid-latitudes, as well as some delayed response in central Africa and South America. Sensitivity simulations (fixing fields affecting transport patterns, chemical kinetics, and biomass burning emissions jointly or separately in GEOS-Chem for the El Niño and La Niña conditions) reveal that the combined effects of transport, chemistry, and biomass burning emissions explain 94%-98% of the simulated variability of TCO in tropical Pacific during ENSO. Changes in transport patterns dominate the overall ozone-ENSO response, by increasing TCO by 0.8 DU (53% of the total variability) in western Pacific region and decreasing TCO by 2.2 DU (92%) in the eastern Pacific region during the El Niño condition relative to the normal periods. Changes in atmospheric temperature, water vapor, and cloud cover enhance ozone chemical depletion, and reduce TCO by 0.2 and 0.7 DU in the WP and EP, respectively. Biomass burning emissions cause an averaged ozone increase of 0.4 and 0.1 DU in respectively for WP and EP region during El Niño, and by 0.7 DU in Brazil during La Niña. The mechanism of the ozone-ENSO responses is summarized in Figure 12.

**Figure 12.** Schematic diagram summarizing the process contributing to the ozone-ENSO response. The underlying figure shows the simulated changes in tropospheric ozone column (TCO) in El Niño, also well as the contribution from three processes (transport, chemistry,

and biomass burning emissions) in the western and eastern Pacific. The numbers are estimated from sensitivity simulations by GEOS-Chem during El Niño conditions (Table 3).

in capturing the ozone-ENSO response. However, we find that five out of ten CCMs in CMIP6 demonstrate good performance in reproducing the ozone-ENSO response, as indicated by their consistent regional mean  $r_{\text{TCO-Niño34}}$  values over the EP and WP regions compared to the results derived from OMI/MLS products. Our results show that the prerequisite for capturing the ozone-ENSO response in CCMs is the inclusion of interactive tropospheric chemistry and the accurate representation of vertical circulation during different ENSO phases. These models further show that the regional mean  $|r_{\text{TCO-Niño34}}|$  and  $|m_{\text{TCO-Niño34}}|$  will increase by  $22\pm9\%$  and  $24\pm4\%$  in the EP regions, and by  $40\pm38\%$  and  $15\pm30\%$  in the WP region, respectively, from the present-day level to future under the SSP3-7.0 scenario. This finding suggests that, given the same magnitude of SST anomalies, ENSO will lead to greater ozone variability in the future, which indicates a stronger response of ozone to ENSO. The stronger ozone-ENSO response is driven by the strengthening anomalous circulation in ENSO phases and more abundant water vapor concentration in a warming climate.

We highlight a number of limitations in this study. First, we do not provide a process-level analyses of the mechanisms contributing to the delayed ozone responses in the mid-latitudes. These responses are weaker compared to the instantaneous response over tropics and may a more complex influence from multiple processes. The ability of the model to capture surface ozone pollution episode caused by the ENSO events are not analyzed. Second, our sensitivity simulations by fixing certain meteorological fields in GEOS-Chem in different ENSO phases may not be able to quantify non-linear impact between meteorology, chemistry, and emissions. In addition, some processes the role of lightning NO<sub>X</sub> emissions are not quantified. This issue may be partly addressed by developing the tagged tracer techniques in the model.

In conclusion, our results reveal the dominant role of atmospheric transport in shaping the overall ozone-ENSO response. They also demonstrate that current CCMs, which are indispensable tools for future projections of atmospheric chemistry and air quality, can successfully reproduce this response when interactive tropospheric chemistry is included and the atmospheric circulation responds accurately to ENSO phases. Ongoing research on the causes and evolution of the ozone-ENSO response is expected to further improve our understanding of the interaction between atmospheric chemistry and climate.

#### Data availability

650

The observational data used in this study is open-access as described in the study. Data from GEOS-Chem modeling that support the findings of this study can be accessed by contacting the corresponding authors (Xiao Lu, luxiao25@mail.sysu.edu.cn; Qi Fan, eesfq@mail.sysu.edu.cn).

# Acknowledgement

This study is supported by the Guangdong Basic and Applied Basic Research Foundation (2025B1515020034, 2024A1515011965), the National Key Research and Development Program of China (2024YFC3714200), and the National Natural Science Foundation of China (42375092). We would like to thank Dr. Jerry R. Ziemke for providing the high-quality OMI/MLS product.

# **Author contribution**

X.L. designed the study. X.L. and Q.F. supervised the project. J.Y.L. performed the model simulations and data analyses with significant input from H.L.W.. J.Y.L. and X.L. wrote the manuscript.

# 665 Competing interests.

The contact author has declared that none of the authors has any competing interests.

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
