# Peer review of "Tropospheric ozone responses to the El Niño-Southern Oscillation (ENSO): quantification of individual processes and future projections from multiple chemical models"

_EGUsphere, 2025_

## Author Comment (AC1)

**Reviewer #2**

**Comment [2-1]:** Li et al. present an interesting analysis of tropospheric ozone responses to ENSO, including a quantification of the effects of transport, chemistry and biomass burning emissions on ozone in the tropical Pacific. The authors utilise the GEOS-Chem model alongside satellite observations for analysis of 'present-day' conditions, as well as CMIP6 models to study projections of future ENSO-ozone relationships.

The scientific questions addressed fall well within the scope of ACP and I recommend publication after the following comments, alongside concerns raised by reviewer 1, are addressed.

**Response [2-1]: We thank the reviewer for the positive and valuable comments. All of them have been implemented in the revised manuscript. Please see our itemized responses below.**

**Comment [2-2]:** To make the research reproducible, more information on the methodology needs to be provided. Reviewer 1 has already raised the question of how future ENSO events are identified in the CMIP6 models. In addition, please specify what correlation and linear regression methods were used to calculate the respective coefficients.

**Response [2-2]: Thank you for your suggestion. We have specified the calculation of the Niño3.4 index in Section2.2: "This index is calculated as:**

$$Niño3.4\ Index = SST - \overline{SST}_{1981-2010} \qquad (1)$$

**where $SST$ is the monthly mean SST averaged over the Niño3.4 region (5°N–5°S, 170°W–120°W), $\overline{SST}_{1981-2010}$ is the 1981–2010 climatological mean SST for the same month over the Niño3.4 region."**

**We have also clarified the calculation of $r_{TCO\text{-}Niño34}$ and $m_{TCO\text{-}Niño34}$ in Section2.5:**

"**The $r_{\text{TCO-Niño34}}$ and $m_{\text{TCO-Niño34}}$ for each grid are calculated as:**

$$r_{X-Y} = \frac{\sum_{i=1}^{n}(X_i - \overline{X})(Y_i - \overline{Y})}{\sqrt{\sum_{i=1}^{n}(X_i - \overline{X})^2}\sqrt{\sum_{i=1}^{n}(Y_i - \overline{Y})^2}} \qquad (2)$$

$$m_{X-Y} = \frac{\sum_{i=1}^{n}(X_i - \overline{X})(Y_i - \overline{Y})}{\sum_{i=1}^{n}(Y_i - \overline{Y})^2} \qquad (3)$$

**Where $X_i$ is the gridded monthly deseasonalised and detrended TCO, $Y_i$ is the monthly Niño3.4 index. These metrics effectively normalize comparisons across models with differing climate variability backgrounds."**

**Comment [2-3]:** The uncertainty in the OMI/MLS retrieval should be introduced in the methodology or discussed when evaluating GEOS-Chem.

**Response [2-3]: Thank you for your suggestion. We have added this information to Section 2.1: "Previous study shows an excellent agreement between the TCO from OMI/MLS and those observed from ozonesonde, with a relatively small bias of about 5 DU. These discrepancies mainly arise from from stratosphere-troposphere separation errors (3-5 DU), and cloud contamination (~2 DU) mitigated by filtering scenes with reflectivity >0.3 (Ziemke et al., 2006)."**

**Reference:**

Ziemke, J. R., Chandra, S., Duncan, B. N., Froidevaux, L., Bhartia, P. K., Levelt, P. F., and Waters, J. W.: Tropospheric ozone determined from Aura OMI and MLS: Evaluation of measurements and comparison with the Global Modeling Initiative's Chemical Transport Model, Journal of Geophysical Research: Atmospheres, 111, https://doi.org/10.1029/2006JD007089, 2006.

**Comment [2-4]:** Explain the choice of boundaries for the west and east Pacific regions in more detail (line 246-247). Were any particular thresholds used or, maybe, do these

regions align with previous studies?

**Response [2-4]: Thank you for pointing it out. We carefully selected the boundaries for the west Pacific (WP) and east Pacific (EP) regions based on two primary considerations: (1) consistency with the established Ozone ENSO Index (OEI) proposed by Ziemke et al. (2010), and (2) statistical robustness within our specific dataset. While our regional definitions are conceptually similar to Ziemke et al.'s OEI, we optimized the boundaries to ensure that approximately 90% of the grid points within the regions shows consistent rTCO-Niño34 with a low p-value.**

**We added the explanation to Section 3.1: "The selection of these regions is based on consistency with the Ozone ENSO Index (OEI) framework (Ziemke et al., 2010), which also ensures that ~90% of the grid points within the regions shows consistent $r_{TCO-Niño34}$ with a low p-value."**

**Reference:**

Ziemke, J. R., Chandra, S., Oman, L. D., and Bhartia, P. K.: A new ENSO index derived from satellite measurements of column ozone, Atmospheric Chemistry and Physics, 10, 3711–3721, https://doi.org/10.5194/acp-10-3711-2010, 2010.

**Comment [2-5]:** Three years of data are used for the El Niño and La Niña conditions to minimize potential impacts from other climate modes. This is not the case for the 'normal' conditions, which are based just on 2013. Please clarify how other climate modes might impact the 'normal' input data or why this is unlikely to be an issue.

**Response [2-5]: Thank you for pointing it out. We agree that using multi-year data to define the "normal" condition is preferable to enhance robustness of analysis, but this would further increase the computational cost as we already have 24 sensitivity experiments on global scale. We select the year 2013 as the reference year based on careful evaluation of index indicating major climate model. In**

addition to theNiño3.4 index analyzed in the text, we also find that the Indian Ocean Dipole (IOD) index stayed below the ±0.5°C threshold and the annual-mean Arctic Oscillation (AO) index is close to zero, indicating this year is not suffered from significant climate variability. Thus, 2013 represents a suitable reference year for "normal" conditions.

We have added the following explanation to Section 2.3: "To reduce computational cost, we only choose year 2013 as the reference year for the Normal period. Evaluation of other climate indexes such as the Indian Ocean Dipole (IOD) index and the Arctic Oscillation (AO) index shows that the year 2013 is not suffered from significant climate variability."

Reference:
https://psl.noaa.gov/data/timeseries/month/data/dmi.had.long.data
https://psl.noaa.gov/data/timeseries/month/data/ao.long.data

**Comment [2-6]:** The 1997/98 El Niño event is discussed in some detail on line 401. I think a brief introduction to the event should accompany the first mention of 1997 ozone levels on lines 53-54.

**Response [2-6]: We have followed your suggestion by adding the text below in Introduction: "A large response of up to 25 DU in tropospheric column ozone was observed over Indonesia during September–November 1997, the period experiencing exceptionally strong El Niño conditions and extreme fires and weather around the world (Page et al., 2002; Picaut et al., 2002), which is comparable to the annual mean level of local tropospheric ozone column (Ziemke and Chandra, 2003)."**

Reference:
Page, S. E., Siegert, F., Rieley, J. O., Boehm, H.-D. V., Jaya, A., and Limin, S.: The

amount of carbon released from peat and forest fires in Indonesia during 1997, Nature, 420, 61–65, https://doi.org/10.1038/nature01131, 2002.

Picaut, J., Hackert, E., Busalacchi, A. J., Murtugudde, R., and Lagerloef, G. S. E.: Mechanisms of the 1997–1998 El Niño–La Niña, as inferred from space-based observations, Journal of Geophysical Research: Oceans, 107, 5-1-5–18, https://doi.org/10.1029/2001JC000850, 2002.

Ziemke, J. R. and Chandra, S.: La Nina and El Nino—induced variabilities of ozone in the tropical lower atmosphere during 1970–2001, Geophysical Research Letters, 30, https://doi.org/10.1029/2002GL016387, 2003.

**Comment [2-7]:** On figure 7, from what I can see, not all the models feature on all the subplots. Please explain whether the missing model values are beyond the scales or whether a particular variable was not available. Additionally, a brief explanation of the Taylor diagram could be included in the figure caption to help the reader. For example, clarifying the axes.

**Response [2-7]: Thank you for your suggestion. We added the explanation to Figure 7 caption: "For Panels (a) and (e), the reference $r_{\text{TCO-Niño34}}$ is from the OMI/MLS observations. For other panels, the reference data is the MERRA-2 reanalysis fields used from GEOS-Chem simulation. The radian axis represents the spatial correlation coefficient between model result and reference data, the X-axis and Y-axis represent the standard deviation (normalized to observations). The distance between a model marker and the reference point ("REF") quantifies the root-mean-square error (RMSE). Thus, markers closer to "REF" indicate better overall performance. By design, the result would not be presented if the simulated values show negative correlation coefficient with the reference data."**

**Comment [2-8]:** On line 182 you introduce the ensemble member for the CMIP6 models and identify a different one is used for UKESM1-0-LL. What is the potential impact of using a different ensemble member on the results? The explanation of

'r1i1p1f1' may provide too much detail if there are no substantial implications of using that particular ensemble members. If there are implications, please highlight them.

**Response [2-8]: Thank you for pointing it out. We selected "r1i1p1f1" as the primary ensemble member to maximize the number of models in our multi-model ensemble, following the common practice in CMIP6 model analyses (Wang et al., 2022). According to the UKESM documentation, the difference between "f1" and "f2" versions is that historical stratospheric aerosol properties are updated in v6.2.0 (f2) to remove errors in some years. This change does not significantly affect ozone simulations or other key variables analyzed in our study (e.g., SST, large-scale circulation). The use of these ensemble members is consistent with other studies using CMIP6 output for ozone analyses (Skeie et al., 2020; Sun and Archibald, 2021). We prefer to provide this information to help the readers to reproduce the results.**

**Reference:**

Skeie, R. B., Myhre, G., Hodnebrog, Ø., Cameron-Smith, P. J., Deushi, M., Hegglin, M. I., Horowitz, L. W., Kramer, R. J., Michou, M., Mills, M. J., Olivié, D. J. L., Connor, F. M. O., Paynter, D., Samset, B. H., Sellar, A., Shindell, D., Takemura, T., Tilmes, S., and Wu, T.: Historical total ozone radiative forcing derived from CMIP6 simulations, npj Clim Atmos Sci, 3, 32, https://doi.org/10.1038/s41612-020-00131-0, 2020.

Sun, Z. and Archibald, A. T.: Multi-stage ensemble-learning-based model fusion for surface ozone simulations: A focus on CMIP6 models, Environmental Science and Ecotechnology, 8, 100124, https://doi.org/10.1016/j.ese.2021.100124, 2021.

Wang, H., Lu, X., Jacob, D. J., Cooper, O. R., Chang, K.-L., Li, K., Gao, M., Liu, Y., Sheng, B., Wu, K., Wu, T., Zhang, J., Sauvage, B., Nédélec, P., Blot, R., and Fan, S.: Global tropospheric ozone trends, attributions, and radiative impacts in 1995–2017: an integrated analysis using aircraft (IAGOS) observations, ozonesonde, and multi-decadal chemical model simulations, Atmos. Chem. Phys., 22, 13753–

13782, https://doi.org/10.5194/acp-22-13753-2022, 2022.

https://ukesm.ac.uk/cmip6/variant-id/

**Comment [2-9]:**

Adding an explanation to the figure captions of the boxes highlighting specific regions in some of the figures (e.g., Fig. 2 g) would provide more clarity.

There are missing 'the', 'a', and 'an' articles throughout the text, e.g. line 15: "Here, we evaluate the GEOS-Chem model…". I assume these will be addressed during the copy-editing stages.

Similarly, occasionally there are mistakes associated with the plural or singular. For example, on line 59: "Mechanisms contributing to the ozone-ENSO response has been examined…". I again assume these will be addressed during the copy-editing stages.

Line 47: "featured by" is unnecessary in the sentence starting "The key response is"

Line 186: should 'forces' be 'forcings' in this sentence?

Lines 190 and 192: rephrase awkward phrasing of "perform interactively tropospheric chemistry" and "perform interactively stratospheric ozone chemistry" to improve readability. For example, change to "simulate tropospheric chemistry interactively".

Line 247: "showing a significant but contrasting ozone response"

Lines 288-290. I suggest splitting in two the sentence starting "The simulated regional mean" for better readability.

Line 324: Remove the unnecessary "by" in "estimated by from the corresponding sensitivity experiments" in the figure caption.

As you state you are not using a threshold for significance, I suggest changing the wording on line 375 from significant to substantial.

I suggest moving the text on limitations (Lines 591 – 597) to earlier in the Discussion and Conclusions section, so that you have a stronger ending focusing on the key results and their implications.

**Response [2-9]: Thank you for your suggestion. We have corrected them accordingly.**

---

## Author Comment (AC2)

**Reviewer #1**

**Comment [1-1]:** This study investigates the response of tropospheric ozone to ENSO using a combination of satellite data, the GEOS-Chem chemical transport model, and CMIP6 chemistry-climate models (CCMs). The authors evaluate GEOS-Chem against OMI/MLS satellite observations, conduct sensitivity experiments to disentangle the roles of transport, chemistry, and biomass burning, and assess how well CMIP6 models capture the observed ozone-ENSO relationship. Finally, the study examines projections under the SSP3-7.0 scenario using selected CMIP6 models.

The key conclusions are:

- GEOS-Chem reproduces observed ozone-ENSO variability very well.
- ENSO-driven changes in transport (via the Walker Circulation) explain most of the ozone variability, though chemistry and biomass burning also contribute.
- CMIP6 models with interactive chemistry capture the ozone-ENSO response more realistically than those with prescribed chemistry.

This is an interesting and timely study that falls well within the scope of ACP. I recommend publication after the following concerns are addressed.

**Response [1-1]: We thank the reviewer for the positive and valuable comments. All of them have been implemented in the revised manuscript. Please see our itemized responses below.**

**Comment [1-2]:** The manuscript would benefit from a deeper discussion of the limitations of the sensitivity experiment design. The assumption of linear additivity may not fully capture the interactions between transport, chemistry, and emissions. For example, transport changes also affect precursor distributions, which in turn influence ozone chemistry. Can the authors quantify how much of the total ozone response is not explained by the sum of the isolated processes (e.g., residuals)? This would help assess

the robustness of the attribution.

**Response [1-2]: Thank you for your suggestion. We have conducted additional analysis to evaluate and discuss the degree of nonlinearity. We derived the interactive effect as the difference between the combined effect (estimated by the TOTAL simulation) and the additive effects of transport, chemistry, and biomass burning emissions, as shown in the revised Table 3. We have added the following paragraph to discuss the interactive effect and the limitation of sensitivity experiment design in Section 3.2: "We can quantify the interactive effect between the chemistry, transport, and biomass burning emissions as the TCO difference between the combined effect and the additive values from the individual effects. Results are shown in Table 3. We find that for the WP region during the El Niño period and the EP region during the La Niña period, the interactive effect tends to amplify the ozone increase. Conversely, for the WP region during La Niña period and the EP region during the El Niño period, the interactive effect tends to weaken the ozone decrease. Part of this interactive effect can be clearly illustrated, as can be seen from the comparison of our result to Sekiya and Sudo (2012) as discussed above. For example, during the El Niño period, higher surface temperature over Indonesia due to the anomalous subsidence may further amplify ozone production from biomass burning emissions, thus the interactive effect leads to a further ozone increase. However, quantifying each interactive mechanism requires much more additional model experiments with more complicated design. Nevertheless, the above analysis again highlights the complex interaction between natural sources, chemistry, and transport in the ozone response to climate variability (Lu et al., 2019a)."**

**Table 3. TCO changes due to the combined and individual effects of transport, chemistry, biomass burning emissions and interactive effect.**

| TCO difference | El Niño | La Niña |
| --- | --- | --- |

| [DU][a] | WP | EP | WP | EP |
|---|---|---|---|---|
| BASE simulation | 1.6 | -2.4 | -0.7 | 0.9 |
| Combined effect | 1.5 | -2.4 | -0.5 | 1.1 |
| Transport | 0.8 | -2.2 | -0.6 | 0.8 |
| Chemistry | -0.2 | -0.7 | -0.6 | -0.2 |
| Biomass burning emissions | 0.4 | 0.1 | 0.1 | 0.1 |
| Interactive effect [b] | 0.5 | 0.4 | 0.6 | 0.4 |

[a] Values are estimated by contrasting the model results using El Niño/La Niña conditions with the Normal periods over the WP and EP region from the sensitivity simulations.

[b] The interactive effect is derived as the difference between the combined effect and the additive effect of transport, chemistry, and biomass burning emissions.

**Comment [1-3]:** The discussion of chemical contributions to the ozone-ENSO response is somewhat limited. It would be helpful if the authors could provide quantitative changes in lightning $NO_X$ and BVOC emissions under ENSO conditions from their simulations. Can these changes be linked to the observed or modelled ozone responses, particularly in the eastern Pacific?

**Response [1-3]: Thank you for pointing it out. We have added the following discussion to Section 3.2: "In comparison, ozone changes induced by lightning $NO_X$ and BVOCs (e.g., isoprene) in response to temperature variations are smaller than those driven by water vapor in the CHEMISTY experiment. Our model simulation yields a small decrease in lightning $NO_X$ emissions of 6% an increase in biogenic isoprene emissions of 4% during the El Niño conditions in the WP region. In the EP region, both lightning $NO_X$ and biogenic isoprene emissions**

**exhibit negligible changes. Thus, the chemistry effect is dominated by changes in water vapor, especially on the global scale.”**

**Comment [1-4]:** While spatial correlation is an informative metric, the authors do not assess how well the models capture the magnitude of interannual variability in TCO. A model may simulate the correct spatial pattern but still underestimate variability. Consider including an evaluation of the standard deviation or amplitude of the TCO–ENSO relationship (e.g., variance in the regression residuals) for each model.

**Response [1-4]: We agree. For the exact reason, we have derived the regression slope $m_{TCO\text{-}Niño34}$ (unit: DU K$^{-1}$) to quantify the magnitude of the TCO change in response to a 1K change in the Niño3.4 index. The comparison between observed and simulated (from GEOS-Chem and CMIP6 models) spatial distributions of $m_{TCO\text{-}Niño34}$ are shown in Figure S2 and Figure S7.**

**We have discussed the ability of the models in capturing $m_{TCO\text{-}Niño34}$ in section 3.1: “The simulated TCO-ENSO sensitivities ($m_{TCO\text{-}Niño34}$) are 1.2 and -1.5 DU K$^{-1}$, which also agree well with the observed values of 1.3 and -1.3 DU K$^{-1}$.”**

**We have also added the following evaluation in section 4.1: “These five models show $m_{mean}$ of 1.0$\pm$0.4 DU K$^{-1}$ for the WP region and -1.4$\pm$0.3 DU K$^{-1}$ for the EP region, compared to the observed values of 1.3 and -1.4 DU K$^{-1}$, respectively.”**

**Comment [1-5]:** The manuscript lacks a clear explanation of how ENSO events are identified in the CMIP6 models under the SSP3-7.0 scenario. Since these models are free-running, ENSO phasing and intensity are not aligned with observations and may differ significantly between models.

**Response [1-5]: Thank you for pointing it out. The ENSO events in the CMIP6 models are identified based on the simulated sea surface temperature (SST)**

averaged over the Niño3.4 region (5°N–5°S, 170°W–120°W) from each model, which is a commonly used method for CMIP models in ENSO research (*e.g.,* Callahan et al., 2021; Cai et al., 2022). The El Niño and La Niña periods in the CMIP6 models are defined following the NOAA standard, with the Niño3.4 index greater than 0.5 and less than -0.5.

We have clarified this point in Section 2.4: "As the CMIP6 simulations analyzed here are not constrained by observed SSTs, the ENSO phases and intensity are not aligned with observations and may differ significantly between models. Following other ENSO-related studies using CMIP6 projections (*e.g.*, Callahan et al., 2021; Cai et al., 2022), we calculate the Niño3.4 index by using the same methodology as Formula 1 but with simulated sea surface temperature (SST) for each CMIP6 model. The Niño3.4 index under future SSP3-7.0 scenario is linearly detrended over 2066-2100. The El Niño and La Niña periods in the CMIP6 models are defined following the NOAA standard, with the Niño3.4 index greater than 0.5 and less than -0.5, respectively."

**Reference:**

Cai, W., Ng, B., Wang, G., Santoso, A., Wu, L., and Yang, K.: Increased ENSO sea surface temperature variability under four IPCC emission scenarios, Nat. Clim. Chang., 12, 228–231, https://doi.org/10.1038/s41558-022-01282-z, 2022.

Callahan, C. W., Chen, C., Rugenstein, M., Bloch-Johnson, J., Yang, S., and Moyer, E. J.: Robust decrease in El Niño/Southern Oscillation amplitude under long-term warming, Nat. Clim. Chang., 11, 752–757, https://doi.org/10.1038/s41558-021-01099-2, 2021.

**Comment [1-6]:** The introduction would benefit from additional references, especially in lines 32, 33, 44, and 46. In particular, the discussion of BVOC and lightning NOx responses to ENSO could be expanded. Suggested references:

https://agupubs.onlinelibrary.wiley.com/doi/full/10.1002/jgrd.50857

https://bg.copernicus.org/articles/20/4391/2023/

https://www.frontiersin.org/articles/10.3389/ffgc.2018.00012/full

**Response [1-6]: We have added the corresponding references and the discussion of BVOC and lightning NO$_X$ responses to ENSO to Section 1: "ENSO also modulates tropospheric ozone concentrations by altering tropic lightning NO$_X$ emissions (Murray et al., 2013), biogenic volatile organic compounds (BVOCs) emissions (Pfannerstill et al., 2018; Vella et al., 2023) and stratospheric-tropospheric exchanges (Doherty et al., 2006; Zeng and Pyle, 2005)."**

**Reference:**

Doherty, R. M., Stevenson, D. S., Johnson, C. E., Collins, W. J., and Sanderson, M. G.: Tropospheric ozone and El Niño–Southern Oscillation: Influence of atmospheric dynamics, biomass burning emissions, and future climate change, Journal of Geophysical Research: Atmospheres, 111, https://doi.org/10.1029/2005JD006849, 2006.

Murray, L. T., Logan, J. A., and Jacob, D. J.: Interannual variability in tropical tropospheric ozone and OH: The role of lightning, Journal of Geophysical Research: Atmospheres, 118, 11,468-11,480, https://doi.org/10.1002/jgrd.50857, 2013.

Pfannerstill, E. Y., Nölscher, A. C., Yáñez-Serrano, A. M., Bourtsoukidis, E., Keßel, S., Janssen, R. H. H., Tsokankunku, A., Wolff, S., Sörgel, M., Sá, M. O., Araújo, A., Walter, D., Lavrič, J., Dias-Júnior, C. Q., Kesselmeier, J., and Williams, J.: Total OH Reactivity Changes Over the Amazon Rainforest During an El Niño Event, Front. For. Glob. Change, 1, https://doi.org/10.3389/ffgc.2018.00012, 2018.

Vella, R., Pozzer, A., Forrest, M., Lelieveld, J., Hickler, T., and Tost, H.: Changes in biogenic volatile organic compound emissions in response to the El Niño–Southern Oscillation, Biogeosciences, 20, 4391–4412, https://doi.org/10.5194/bg-20-4391-2023, 2023.

Zeng, G. and Pyle, J. A.: Influence of El Niño Southern Oscillation on

stratosphere/troposphere exchange and the global tropospheric ozone budget, Geophysical Research Letters, 32, https://doi.org/10.1029/2004GL021353, 2005.

**Comment [1-7]:** The SST values used in the sensitivity simulations should be described more clearly.

**Response [1-7]: Thank you for your suggestion. In our sensitivity simulations using offline GEOS-Chem model, the model was driven by MERRA-2 reanalysis meteorological fields rather than direct SST inputs, same as in the BASE simulation. MERRA-2 reanalysis meteorology provides fully consistent, observationally constrained atmospheric states (including derived SST influences on atmospheric processes) that better represent real-world conditions.**

**Comment [1-8]:** More explanation is needed on how r_TCO–Niño3.4 is calculated. Are the Niño3.4 index values spatially uniform?

**Response [1-8]: Thank you for your suggestion. The Niño3.4 index is spatially uniform. We have clarified in Section 2.5: "The $r_{\text{TCO-Niño34}}$ and $m_{\text{TCO-Niño34}}$ for each grid are calculated as:**

$$r_{X-Y} = \frac{\sum_{i=1}^{n}(X_i - \overline{X})(Y_i - \overline{Y})}{\sqrt{\sum_{i=1}^{n}(X_i - \overline{X})^2}\sqrt{\sum_{i=1}^{n}(Y_i - \overline{Y})^2}} \tag{2}$$

$$m_{X-Y} = \frac{\sum_{i=1}^{n}(X_i - \overline{X})(Y_i - \overline{Y})}{\sum_{i=1}^{n}(Y_i - \overline{Y})^2} \tag{3}$$

**Where $X_i$ is the gridded monthly deseasonalised and detrended TCO, $Y_i$ is the monthly Niño3.4 index. These metrics effectively normalize comparisons across models with differing climate variability backgrounds."**

**Comment [1-9]:** While the manuscript avoids using a p-value threshold, some

discussion of statistical confidence is warranted. How confident are the authors that the reported correlations and sensitivities exceed internal variability?

**Response [1-9]: Thank you for raising this point. We continue to use the p-value as a valuable metric for quantifying statistical confidence, but avoid using thresholds such as p $\leqslant$ 0.05 to judge whether the reported values are statistically "significant". We clarify in Section 2.5: "We report the *p*-value of corresponding *r*TCO-Niño34 and *m*TCO-Niño34 where applicable, but we do not use thresholds such as $p \leqslant 0.05$ to judge whether the reported values are statistically significant, as advised by the statistics community (Wasserstein et al., 2019). Still, smaller p- value indicates higher statistical confidence."**

**Reference:**

Wasserstein, R. L., Schirm, A. L., and Lazar, N. A.: Moving to a World Beyond "p < 0.05," The American Statistician, 73, 1–19, https://doi.org/10.1080/00031305.2019.1583913, 2019.

**Comment [1-10]:** The manuscript would benefit from a brief overview of the SST and ocean components in the CMIP6 models.

**Response [1-10]: We have added the information of the ocean component of CMIP6 in Table S1 and briefly introduced it in Section 2.4: "Table S1 summarizes the ocean components of the CMIP6 models analyzed in this study, including their resolutions. These model configurations represent the current generation of ocean-atmosphere coupling systems used for simulating ENSO dynamics."**

**Table S1. Ocean components and sea surface temperature SST calculation information of the CMIP6 models used in this study.**

| Name | Ocean components | Resolution | Reference |
|---|---|---|---|
| AWI-ESM-1-1-LR | FESOM 1.4 | 50km | Shi et al. (2020) |
| BCC-ESM1 | MOM4 | 50km | Wu et al. (2020) |
| CESM2-WACCM | POP2 | 100km | Danabasoglu et al. (2020) |
| EC-Earth3-AerChem | NEM3.6 | 100km | Döscher et al. (2022) |
| GFDL-ESM4 | MOM6 | 25km | Dunne et al. (2020) |
| IPSL-CM6A-LR-INCA | NEMO-OPA | 100km | Boucher et al. (2020) |
| MPI-ESM-1-2-HAM | MPIOM1.63 | 50km | Mauritsen et al. (2019) |
| MRI-ESM2-0 | COM4.4 | 100 km | Yukimoto et al. (2019) |
| NorESM2-MM | MICOM | 100 km | Seland et al. (2020) |
| UKESM1-0-LL | NEMO-HadGEM3-GO6.0 | 100 km | Sellar et al. (2019) |

**Comment [1-11]:**

The frequent use of opposing effects in parentheses (e.g., "increase (decrease)") in the abstract and main text is hard to read. Consider rephrasing for clarity.

**Response [1-11]: We have revised where applicable. However, due to word limit, this usage has been retained in the abstract.**

**Comment [1-12]:**

Lines 136–137 suggest that GEOS-Chem runs freely, but the model is in fact driven by nudged reanalysis meteorology. Please clarify this to avoid contradiction.

Line 205 – consider rephrasing to improve clarity.

Line 274 – citation needed.

Line 364 – "nudging" is more accurate than "imposing."

Line 375 – are these effects statistically significant?

Line 399 – citation needed.

Use the more established term Chemistry-Climate Models (CCMs) instead of "climate-chemistry models."

Line 374 – contains a typo.

Lines 510–517: The explanation of future projections is unclear. How are you comparing responses under "the same SST anomalies" when SSTs are not synchronised across free-running models? Please clarify or rephrase.

**Response [1-12]: Thank you for pointing it out. We have corrected them accordingly.**